# Self-Reported Health Problems and Quality of Life in a Sample of Colombian Childhood Cancer Survivors: A Descriptive Cross-Sectional Study

**DOI:** 10.3390/cancers14122999

**Published:** 2022-06-18

**Authors:** Natalia Godoy-Casasbuenas, Esther de Vries

**Affiliations:** 1PhD Program in Clinical Epidemiology, Department of Clinical Epidemiology and Biostatistics, Faculty of Medicine, Pontificia Universidad Javeriana, Bogotá 110231, Colombia; 2Department of Clinical Epidemiology and Biostatistics, Pontificia Universidad Javeriana, Bogotá 110231, Colombia; estherdevries@javeriana.edu.co

**Keywords:** survivors of childhood cancer, self-reported health problems, quality of life

## Abstract

**Simple Summary:**

Childhood cancer survivors are currently an understudied population in Colombia and, in general, in South America. Indeed, the attention in this region is still largely on curative care for childhood cancers, and the group of childhood cancer survivors is not a focus point; there are no descriptions of the presence of adverse events that may have presented in the short, medium, or long term in this population This article in an observational, descriptive cross-sectional study of 122 Colombian childhood survivors who were invited to complete a self-reported study-specific online questionnaire along with the SF-36 Health Survey. The aim of this study is to describe the perceived long-term health problems and quality of life among Colombian adults who had cancer in their childhood or adolescence. This is, therefore, a first step to characterize this population and as an input for the formulation of long-term follow-up goals.

**Abstract:**

Objectives: To describe the self-reported health problems and quality of life among adult-aged Colombian childhood and adolescent cancer survivors. Methods: This is a descriptive cross-sectional study with Colombian childhood cancer survivors (CCS) who were diagnosed before the age of 18, at the moment of study were ≥18 years, and at least 5 years had passed since diagnosis. Each participant completed a self-reported study-specific online questionnaire along with the SF-36 Health Survey to assess the prevalence of health problems and current quality of life (QoL). Data were analyzed using descriptive statistics and independent sample *t*-tests. Results: Out of the 122 CCS who participated, 100% reported at least one health problem, mostly gastritis, headaches, and lack of concentration, followed by obesity and fertility issues. In general, they had a good perception of their QoL, which was, on average, only diminished in the areas of vitality, emotional role functioning, and social functioning. Conclusion: Perceived health problems among the participating Colombian CCS were prevalent; most reported a good self-perceived QoL. This is the first study on understanding health problems and QoL of CCS treated in Colombia and South America. It reopens the debate on the need to carry out long-term follow-up in this population among Colombian society.

## 1. Introduction

The treatment of childhood cancer has been one of the great success stories of modern oncology [1,2]. Prior to the 1970s, most children or young adults diagnosed with cancer had little hope of being cured [3]. Since then, cure rates and 5-year survival probabilities have increased dramatically, reaching 80% in developed countries, especially for leukemias [4,5,6]. As a result, the number of CCS has grown dramatically. Along with the impressive gains in survival have come late effects of the cancer and its treatment, affecting the health and quality of life (QoL) of some survivors [6]: it is estimated that two thirds of these CCS will experience a chronic adverse health event [7], leading to considerable morbidity and a risk of premature death [8]. In industrialized countries, the population of CCS has been studied for more than 30 years, showing that they have a higher risk of subsequent cancers, cardiac events, pulmonary conditions, fertility issues, neurocognitive impairment, among others, leading to a reduced quality of life [7,9,10]. Additionally, several psychological effects have also been described as potential consequences that CCS can develop, including depression, anxiety, poor peer relations, post-traumatic stress disorder [11], loneliness, and low school performance [12,13,14,15,16]. Furthermore, lifestyle choices among this group may be significantly affected by their cancer experience [16]. Due to this variety of adverse events, follow-up strategies have been designed for this population in several high-income countries.

In Colombia, childhood cancer represents around 3% of new cancer cases [17]. Although there has been a progressive increase in overall survival of childhood cancer in Colombia [18], due to the implementation of multimodal therapeutic regimes and health policies aimed at reducing treatment abandonment [17,19], cancer still represents the second highest cause of childhood mortality. Survival differs substantially between regions of the country, despite having standard therapeutic regimens. Contributing factors to these differences include the area of residence (urban or rural), administrative barriers to therapeutic access, distance to reach medical facilities, and family support [20,21].

To our knowledge, no characterization of childhood cancer survivors in Colombia or in South America has been performed. The attention in South America is still largely on curative care for childhood cancers, and the group of childhood cancer survivors is not a focus point; there are no descriptions of the presence of adverse events that may have presented in the short, medium, or long term in this population [22]. Follow-up strategies to assess the long-term adverse events in these patients have not yet been addressed in Colombia. Some other Latin American countries began to implement some of these follow-up programs for CCS, such as the PINDA program (Programa Infantil Nacional de Drogas Antineoplásicas), which includes a follow-up program in Chile [23,24,25] or the proposal of the follow-up card for CCS in Mexico [26]. The objective of such long-term follow-up is to facilitate diagnosis in a timely manner, and thus to be able to appropriately manage late adverse events, reducing the frequency of severe complications [3,27,28].

This cross-sectional study aims to describe the self-reported health problems and QoL among adult-aged Colombian childhood and adolescent cancer survivors. The results will serve as a first step in characterizing this population that will support the formulation of long-term follow-up care goals.

## 2. Materials and Methods

### 2.1. Study Design

A descriptive cross-sectional study was used to meet the objective of this study. It was performed in Colombia between February 2021 and January 2022.

In Colombia, there is no registry of childhood cancer survivors. Therefore, cross-linking with data from the few population-based cancer registries to identify survivors is challenging. This is because national ID numbers do not provide contact information and, additionally, up until recently childhood ID numbers were different from adult ID numbers. The absence of such a unified identification number in combination with frequent transferals of patients by their insurance has made long-term follow-up of childhood oncology patients growing into adults almost impossible.

Therefore, we used different non-probabilistic sampling strategies, such as purposeful and snow-ball sampling, to identify Colombian CCS: First, there was an active search of adults with a history of childhood cancer that were contacted by personal contacts of the author, and by adult and pediatric oncologists—who remain in contact with former patients as the cancer experience has such a great impact. Secondly, we contacted several cancer patients’ foundations, who provided lists of CCS who are currently old members of the foundations. This helped us to reach out to invite CCS from different regions of Colombia.

Recruitment was also undertaken using open invitations on social media platforms such as Facebook, Instagram, Twitter, and cancer organizations’ web pages, where we invited Colombian CCS to participate and included the link to the informed consent page and the questionnaire.

Inclusion criteria were the following: (1) adults (aged ≥ 18 years) with a medical history of childhood cancer diagnosis, (2) age < 18 years at the time of diagnosis, (3) having survived at least 5 years since the diagnosis, and (4) agreement to participate through informed consent. We did not include subjects with a diagnosis of non-malignant tumors (i.e., Langerhans cell histiocytosis, meningioma, craniopharyngioma, etc.) treated with radiation and/or chemotherapy, nor subjects who were unable to answer the questionnaire by themselves. There was one retinoblastoma survivor who was blind who participated: the researcher met with her, read out loud all of the questions, and completed the questionnaire according to her answers.

### 2.2. Measures and Questionnaires

Participants who accepted the invitation to participate were directed to an online platform (in REDCap), where first they received written information followed by an electronic informed consent form. Upon accepting to participate and signing informed consent, they were directed to the online questionnaire evaluating socio-demographic characteristics (age, level of education, marital status, employment status, access to health security services, etc.), self-reported cancer medical history, and treatment characteristics. Perceived health problems were assessed using questions like the ones employed in the Long-Term Follow-up Study [29], organized by the different systems (cardiovascular, respiratory, neurologic, etc.) (details in annex1). Medical conditions that participants had ever experienced in their lifetime were assessed with structured questions, such as “Have you ever been told by a doctor or other health care professional that you have or have had (name of the condition) for example: “congestive heart failure?””. The answer options were “Yes”, “No”, “I don´t know” and “If yes, please tell us the age of occurrence”.

The questionnaire also included the SF-36 Health Survey to assess the current QoL of the Colombian CCS. This instrument has been validated for Colombia [30] and consists of 36 items, covering eight domains of health: functional capacity, physical aspects, bodily pain, overall health status, vitality, social aspects, emotional aspects, and mental health. Each question in the SF-36 is given a score that is later translated to a scale from 0 to 100, in which zero corresponds to the worst health status and 100 to the best [30,31].

Finally, the questionnaire included a few open questions where CCS could express their opinion on what kind of information or strategy could be available to improve the care of Colombian CCS.

In all, the questionnaire comprised 381 items, of which 204 dealt with cancer history, surgeries, and medical conditions by systems, 31 dealt with overall health and other concerns, and 47 dealt with QoL issues. The instrument is provided as a Appendix A. The questionnaire was uploaded into the REDCap platform hosted at the San Ignacio Hospital to systematize the answers and facilitate the dissemination of the study.

### 2.3. Statistical Analysis

A descriptive analysis was performed. The socio-demographic characteristics of the participants of the study were described, as well as the percentages of health events according to the different systems (cardiovascular, pulmonary, endocrinological, neurological, ophthalmic, etc.). A descriptive analysis of the SF-36 QoL instrument was also performed. Descriptive statistics are presented as frequencies, mean values, median values, and percentages. Bivariate analysis was performed using Student’s *t*-test or analysis of variance (ANOVA) for the comparison of quantitative variable means. To partially overcome the absence of a control group in this study, published data on QoL using SF-36 coming from the city of Medellín was used to compare the mean scores of the eight health domains of the sample of CCS with those of the general population of this Colombian city [32]. The statistical analyses were performed using STATA version 17 [33].

### 2.4. Ethics

All methods and instruments used during this study were revised and approved by the research and ethics committee of San Ignacio University Hospital of Pontificia Universidad Javeriana (FM-CIE-0169-21). Participants provided an electronically signed informed consent before being able to proceed to the questionnaire.

## 3. Results

Out of 132 questionnaires collected, 122 were valid. Four of the invalid surveys corresponded to participants who partially replied to the questionnaire. Three responses were invalid due to technical difficulties of participants in logging into the REDCap questionnaire, resulting in multiple attempts in answering the survey, in which case only the (most) complete attempt was included. Two surveys were filled out by persons who were diagnosed less than five years prior to the date of participation, and one by a participant who was still under 18 years of age.

### 3.1. Study Population

The final study sample consisted of 122 persons (69 women, 52 men, 1 other gender). The median current age of the participants was 27 years (range: 19–65), most of them were single (71.3%), had achieved technical education (33.6%), and were currently working (54.9%) (Table 1). The majority came from Bogota (71.9%) the capital city, from Pereira (12.6%), and Barranquilla (9%). In total, 24 (19.7%) had children (Table 1).

Concerning their cancer history, the median age at diagnosis was 9.2 years (range: 0–18), the most common cancer was leukemia (43.4%), followed by Hodgkin lymphoma (17.2%) and Wilms tumor (10.6%). The rest of the group included cases of osteosarcoma (4.1%), rhabdomyosarcoma (4.1%), Ewing sarcoma, hepatoblastoma, histiocytosis (1.6%), and there were two cases of CCS with retinoblastoma. The rest of the group, defined as “other diagnosis”, included rare malignant disorders, for example, ovarian tumors or bladder cancer (Table 2). Most were diagnosed 10–20 (36.6%) and 20–30 (37.4%) years ago: the oldest participant (currently 65 years old) was a lymphoma survivor treated in the 1970s. The vast majority were treated with chemotherapy (97.5%), followed by radiotherapy (43.4%) and surgery (39.3%).

Regarding major sequelae at the present time, 57 (46%) participants reported having persistent hair loss, 39 (31.5%) participants had scarring or disfigurement of the chest or abdomen region, 23 (18.6%) had scarring or disfigurement of the arms or legs (including an abnormally short arm or leg), and 8 participants (6.5%) walk with a limp. One osteosarcoma survivor lost her leg, and two retinoblastoma survivors lost an eye, one unilaterally and the other bilateral, which resulted in complete blindness. Thirteen participants (10.7%) mentioned having had a second neoplasm (Table 2).

### 3.2. Participants’ Descriptions of Their Current Health Problems

The results of the descriptive analyses for the reported health problems are summarized in Table 3. Among the most prevalent health problems reported by this sample of 122 Colombian CCS, 40 (32.8%) mentioned having gastritis, 34 (27.9%) had frequent headaches, and 23 (18.9%) lack of concentration and memory. Twenty-three survivors (18.5%) also mentioned having obesity issues and 34 (18.9%) participants mentioned having difficulties in having children. Most of the reported health problems appeared at around the age of 20, except the different neurological manifestations which were reported to appear as of adolescence. A total of 35 (28.7%) participants had been told they may have difficulties having children, of which 16 already had medical exams (blood tests, ultrasounds, sperm count).

Concerning the visits to the different medical specialties, 34 (27.9%) had consulted the cardiologist, 31 (25.4%) a neurologist, 29 (23.8%) a gastroenterologist, and 22 (18.0%) a pulmonologist. In total, 54 (43.9%) participants had not visited any of these specialists. 

Finally, regarding life after cancer, 17 participants mentioned that some of their biggest concerns included having an adverse event due to treatment and 5 participants mentioned fear of relapse and being concerned about inheriting the disease to their offspring and other fertility issues. One participant mentioned that his biggest concern was suffering from employment discrimination based on his medical history.

### 3.3. Participants’ Descriptions of Their Quality of Life

A total of 118 participants answered the SF-36 Health Survey questionnaire, the results of which are shown in Table 4. The scores obtained from the eight domains of the SF-36 revealed that both the physical functioning and physical role functioning had the highest health-related QoL scores. Bodily pain, general health, social functioning, and emotional role functioning were at the mid-point of the scores (around 72). The most affected domains were vitality and mental health (61.5 and 67, respectively). When comparing leukemia CCS with solid tumors CCS, the latter had lower values in the dimensions of physical functioning, bodily pain, general health, social functioning, emotional role functioning, and mental health.

Comparing participant´s QoL with Medellín´s general population for the same sex, it is observed that Colombian CCS obtained a lower score on the dimensions of vitality, social functioning, and emotional role functioning. This difference was significant in the dimension of social role functioning. The general population had a lower score on the dimensions of physical functioning, physical role functioning, bodily pain, and mental health (Table 5).

### 3.4. Participants’ Proposal for Improving Colombian CCS Care

When asking participants about the actions that can be undertaken to improve Colombian CCS care in the open questions, the following aspects were highlighted: (1) Receiving information on the potential late adverse events and warning signs, as well as a copy of the medical history at the end of treatment; (2) Having national and unified guidelines on childhood cancer survivors follow-up; (3) Creating an App with virtual information on post-cancer treatment care issues; (4) Receiving psychological support in the process of “getting back to normal life”; (5) Receiving fertility support; (6) Education in self-care and healthy lifestyle habits.

## 4. Discussion

This study presents data on the prevalence of self-reported health problems and QoL in a sample of Colombian CCS. This prevalence cannot be considered representative due to the sampling method. The majority of survivors mentioned at least one health problem. The most prevalent ones were gastritis, frequent headaches, and a lack of concentration—the latter mostly reported by many brain cancer and leukemia CCS. Several studies conducted in the United States and in Europe have described the neurocognitive problems in these survivors. These complaints are thought to be due to the brain´s vulnerability to the neurotoxicity of the therapies used to treat these types of cancer: cranial radiotherapy, chemotherapy, and surgery [34,35,36]. The other most prevalent conditions were related to being overweight or obese, another common finding among CCS reported in several large CCS cohort studies, where long-term survivors or acute lymphoblastic leukemia (ALL) and those who received abdominal radiotherapy were at the highest risk of obesity and other cardiometabolic conditions [37,38,39,40,41]. Additionally, Colombian CCS from this study mentioned difficulties regarding having children and being worried about vulnerability to cancer in their offspring. Several other studies also indicate that CCS worry about their reproductive capacity and/or the health of offspring [42]. This is particularly relevant for CCS who received treatment during adolescence and are now thinking about having children. Indeed, the exposure of high-dose alkylating chemotherapy and abdominal/pelvic radiotherapy adversely affects gonadal function in CCS [42,43,44,45]. A study carried out in The Netherlands that evaluated educational achievement, employment status, living situation, marital status, and offspring in a large sample of young adult CCS compared with a control group, revealed that both female and male survivors reported worrying significantly more about their fertility than their peers and they also worry often about the health of their future children. They also wonder if they could pass on their cancer genetically to their children [46]. Interestingly, the age of occurrence of the different self-reported health problems among Colombian CCS is quite young compared to other cohort studies. In our study, the median age of occurrence of the different health problems is during late adolescence/early adulthood, while in other CCS cohorts, the age of occurrence is 15 to 20 years after the cancer treatment (around 30–40 years old) [47,48]. This raises a crucial aspect as to the importance of starting surveillance of this population as soon as possible after medical treatment. 

In general, the CCS participating in our study reported a good self-perceived QoL. Compared to the general population, Colombian CCS had significantly lower values only for the dimension of emotional role functioning, but higher significant values for the dimensions of physical functioning, physical role functioning, bodily pain, and mental health. This finding confirms that the QoL is not worse than that of the general population, yet differs from some studies using the SF-36 that have found health-related QoL among CCS to be comparable to that of the general population [49]. For example, a study with Greek CCS reported that survivors’ scores on most subscales of the SF-36 were similar to those of controls, despite some difficulties in their daily activities [50]. Additionally, evidence from the Swiss Childhood Cancer Survivor Study showed that ALL survivors reported a good health-related QoL compared to the general population [51]. In our study, 43% of the sample were survivors of childhood leukemia, implying an expectation of a reasonably good QoL after treatment. However, there is also a large amount of research with the SF-36 in CCS where most of CCS experience worse QoL than the general population in almost all domains [52,53,54,55]. According to van Erp et al, there are several reasons that can explain these conflicting results: 1. The differences in the survivor groups that are included, such as diagnosis or time of follow-up; 2. The use of different reference groups to make these comparisons (siblings, healthy peers, or the general population) [56]. As our study did not include a control population and there is no Colombian study providing normative or “average population” SF-36 values, we used as a comparison group data collected in Medellín, a large Colombian city, which is unfortunately not representative of the Colombian general population.

### 4.1. Strengths and Limitations

To our knowledge, this is the first study to explore long-term health problems and QoL in Colombian CCS as a first attempt in characterizing this population. Thanks to the different sampling strategies, we were able to reach CCS in several cities around the country and not only in the capital city. The help of patient foundations and use of social networking sites helped to recruit hard-to-reach participants as CCS. We were able to identify CCS with a variety of cancer diagnosis, ages (ranging between 19 and 65 years) and prolonged times since diagnosis, allowing us to capture self-reported health problems and QoL that may be different in those patients treated in different time periods.

The main limitations include the lack of a sampling frame, limiting generalizability, and the lack of a control group. The lack of sampling frame and the use of purposeful sampling may have created selection bias. Indeed, some of the survivors who participated in this study are still linked to their oncologist or the cancer foundation they were part of; thus, the sample may reflect more ongoing health problems. This inhibits us from making statements regarding the representativeness of our respondents to the whole population of CCS in Colombia. There was, however, no other way to identify hard-to-reach and hidden populations of adults with a history of childhood cancer currently integrated in the community [57,58]. The lack of a control group implies an absence of frequency of our outcomes in a non-cancer population, preventing direct comparisons with survivors. Other limitations are inherent to the cross-sectional and descriptive nature of our study and the use of self-reported questionnaires. However, several studies have used self-reported questionnaires to assess the presence of late effects and health-related QoL among lymphoma CCS [59]. Furthermore, a study conducted among CCS in Korea using a self-reported questionnaire revealed that perceived health problems were prevalent among CCS and were significant in assessing physical and mental functioning [60]. Finally, we recognize that the study findings do not mention intensity of health problems and may fail to capture all possible health problems, as we listed the most relevant ones.

### 4.2. Implications

Although our findings mainly rely on self-reported measures, this pioneer study on adult survivors of childhood cancer adds knowledge on characterizing the current health status and perceived QoL in this unstudied population in Colombia. The presence of current health problems and a perceived QoL diminished in the areas of vitality and emotional role functioning is a start in supporting the need of establishing follow-up strategies for this population in this country. Currently, most high-income countries have guidelines for childhood cancer survivorship care [27,61,62,63,64]. In Colombia, follow-up care for cancer survivors is starting to be implemented among adult cancer survivors, but there is still a long way to go for implementing these strategies among CCS [65]. For the survivors of childhood cancer of this study, life after cancer also brings its challenges and concerns, such as the perceived risk of developing an adverse event after treatment, having a relapse, and concerns related to fertility. These concerns can also extend to the social sphere with feelings of employment discrimination due to medical history, among others, which sometimes leads to hiding or not mentioning this event. Understanding the impact of perceived health problems and QoL in Colombian adult survivors of childhood cancer, can positively impact in improving the current medical attention of CCS. Additionally, it will allow for the planning of a better transition to the after cancer and “returning to normal life” phase, considering not only their physical needs, but also their mental and psychological ones.

## 5. Conclusions

In this study, self-reported health problems among Colombian CCS were prevalent and although they reported a good perception of their QoL, this study is unique in adding knowledge in understanding CCS treated in Colombia. It reopens the debate on the need to carry out long-term follow-up in this population among Colombian society.

## Figures and Tables

**Table 1 cancers-14-02999-t001:** Socio-demographic characteristics of Colombian CCS.

Characteristics	Cases *n* = 122 (%)
**Gender**	
Male	52 (42.6%)
Female	69 (56.6%)
Other	1 (0.8%)
Age (mean–std)	28.6 (8.1)
**Age, years**	
Missing birth date	1 (0.8%)
18–29	77 (62.6%)
30–39	33 (26.8%)
40–49	10 (8.1%)
>50	1 (0.8%)
**Marital status**	
Single	87 (71.3%)
Married	15 (12.3%)
Free union	17 (13.9%)
Separated	2 (1.6%)
Divorced	1 (0.8%)
**Level of education**	
Without formal education	1 (0.8%)
Primary education or below	33 (27.1%)
High school	30 (24.6%)
Technical education	41 (33.6%)
Undergraduate	15 (12.3%)
Postgraduate studies	2 (1.6%)
**Current occupation**	
Work	67 (54.9%)
Student	43 (35.2%)
Stay-at-home parent	5 (4.1%)
Other	7 (5.7%)
**Offspring**	
No	98 (80.3)
Yes	24 (19.7%)
**Number of children**	
1	13 (56.2%)
2	9 (39.1%)
3	1 (4.4%)
**City**	
Bogotá	88 (71.9%)
Pereira	15 (12.6%)
Barranquilla	11 (9.0%)
Bucaramanga	5 (4.1%)
Cali	1 (0.8%)
Ibague	1 (0.8%)
Cucuta	1 (0.8%)

**Table 2 cancers-14-02999-t002:** Cancer history characteristics of Colombian CCS.

Cancer History Characteristics	Cases (*n* = 122)	Cancer History Characteristics	Cases (*n* = 122)
**Type of childhood cancer**		**Time since diagnosis**	
Leukemia	53 (43.4%)	5–10	20 (16.2%)
Hodgkin lymphoma	21 (17.2%)	10–20	45 (36.6%)
Wilms tumor	13 (10.6%)	20–30	46 (37.4%)
Brain tumor	6 (4.9%)	>30	11 (8.9%)
Osteosarcoma	5 (4.1%)	**Hospitalized due to complications**	Yes 92 (74.2%)
Rhabdomyosarcoma	5 (4.1%)		No 32 (25.8%)
Ewing sarcoma	2 (1.6%)	**Hospitalized in the ICU**	Yes 51 (41.1%)
Histiocytosis	2 (1.6%)		No 62 (50.0%)
Hepatoblastoma	2 (1.6%)		DoR* 11 (8.9%)
Retinoblastoma	2 (1.6%)	**Second neoplasms**	
Other diagnosis *	12 (9.7%)	Yes	13 (10.7%)
**Stage**		No	108 (89.2%)
Early stage	68 (55.7%)	**Major sequelae at the present time:**	
Advanced stage	39 (32.0%)	Persistent hair loss	Yes 57 (46.0%)
Doesn´t know	15 (12.3%)		No 67 (54.0%)
**Age of childhood cancer**		Scarring or disfigurement of the head or neck region	Yes 17 (13.7%)
Age (mean–DS)	9.2 (4.9)		No 107 (86.3%)
0–5	37 (30.3%)	Scarring or disfigurement of the chest or abdomen region	Yes 39 (31.5%)
6–10	34 (27.9%)		No 85 (68.6%)
11–15	38 (31.2%)	Scarring or disfigurement of the arms or legs (including an abnormally short arm or leg)	Yes 23 (18.6%)
16–18	15 (12.3%)		No 101 (81.4%)
**Length of treatment**			
1 year	33 (27.1%)	Walks with a limp	Yes 8 (6.5%)
2 years	41 (33.7%)		No 116 (93.6%)
3 years	18 (14.8%)	Loss of an arm, leg, finger, or toe	Yes 1 (0.8%)
>3 years	30 (24.6%)		No 123 (99.2%)
**Treatment**		Loss of an eye	Yes 2 (1.6%)
Chemotherapy	Yes 119 (97.5%)		No 122 (98.4%)
Radiotherapy	Yes 53 (43.4%)		
Surgery	Yes 48 (39.3%)		

* DnR: Does not respond.

**Table 3 cancers-14-02999-t003:** Major self-reported health problems by Colombian CCS.

Medical Condition	Yes *n* (%)	Age of Occurrence Median (IQR)	Medical Condition	Yes *n* (%)	Age of Occurrence Median (IQR)
**Hearing/vision/speech**			**Reproductive system**		
Hearing loss	5 (4.1%)	17 (16–18)	Difficulty having children	36 (29.0%)	15 (13–17)
Deafness in one or both ears	3 (2.5%)	16 (15–17)	Medical tests to see if you might have problems having children	16 (12.9%)	19.5 (16–29.5)
Tinnitus	7 (5.7%)	17 (15–18)	Men: low sperm count	4 (7.6%)	21.5 (16–28)
Dizziness or persistent vertigo	10 (8.1%)	14 (8–19)	Currently have menstrual period	60 (85.7%)	12 (12–13)
Blindness of one or both eyes	5 (4.0%)	5 (2–15)	**Cardiovascular system**		
Cataracts in one or both eyes	2 (1.6%)	32 (6–58)	Arrhythmias	6 (4.9%)	15 (12–20)
Any vision problems in one or both eyes, even when wearing glasses?	35 (28.3%)	15 (12–20)	Heart failure	4 (3.2%)	14.5 (12–15.5)
Dry eye requiring lubricating drops	15 (12.1%)	17 (15–25)	Hypertension	7 (5.7%)	14 (12–17)
Chewing defect	5 (4.0%)	15 (14.5–18.5)	Angina	7 (5.7%)	16 (13–32)
Phonation defect	3 (2.4%)	7 (4–13)	Shortness of breath or irregular heartbeat while exercising	14 (11.3%)	15.5 (12–20)
**Dental system**					
Malformations in the teeth or jaw	17 (13.7%)	14 (8–15)	**Respiratory system**		
Decrease in the production of saliva	2 (1.6%)	18.5 (17–20)	Asthma	9 (7.0)	9 (7–12.5)
**Skin and annexes**			Recurrent pneumonia	2 (1.6%)	12
Problems or stains on nails	15 (12.1%)	13 (7–16)	Fibrosis	1 (0.8%)	17
Abundant hair loss	31 (25%)	16 (10–20)			
Lack of hair	21 (16.9%)	14.9 (12–20)	**Digestive system**		
**Urinary system**			Hepatitis	10 (8.1%)	14 (9–25)
Repeated kidney infections	6 (4.8%)	14.5 (10–20)	Gastritis	42 (33.9%)	18 (14–25)
Kidney stones	7 (5.7%)	19 (16–25)	Chronic diarrhea	2 (1.6%)	12.5 (10–15)
Repeated urinary infections	14 (11.3%)	16 (13–21)			
			**Brain and central nervous system**		
**Endocrine system**			Frequent headaches	35 (28.2%)	15 (13–18)
Obesity	23 (18.5%)	19 (14–23)	Seizures	10 (8.2%)	12.5 (13–18)
Diabetes	3 (2.5%)	26 (2–61)	Balance problems	8 (6.5%)	5.5 (11.5–18)
Dyslipidemia	5 (4.0%)	27.4 (20–40)	Lack of concentration and memory	24 (19.4%)	15 (12–18)
Taking growth hormone	8 (6.5%)	12 (10–18)	**Medical visits**		
Osteoporosis	3 (2.5%)	15 (11–30)	Visit neurologist	31 (25.4%)	15 (12–18)
			Visit to cardiologist	33 (27.1%)	18.4 (13–20)
			Visit pulmonologist	23 (18.6%)	16.5 (9–19)
			Visit gastroenterologist	30 (24.2%)	11.4 (10–27)

**Table 4 cancers-14-02999-t004:** Description of quality of life based on SF-36 questionnaire according to cancer type.

QoL Dimension	Sample of Colombian CCS (*n* = 118)	Leukemia	Lymphoma	Solid Tumors
		(*n* = 52)	(*n* = 21)	(*n* = 19)
Item	Mean (DS)	Mean (DS)	Mean (DS)	Mean (DS)
Physical functioning	92.1 (13.5)	93.1 (11.7)	95.5 (8.2)	82.6 (21.7)
Physical role functioning	88.1 (27.9)	86.5 (30.3)	90.5 (24.3)	90.8 (23.9)
Bodily pain	75.9 (19.5)	76.5 (18.9)	77.2 (17.1)	73.6 (21.8)
General health	73.1 (19.6)	76.5 (18.6)	65.6 (18.7)	68.2 (19.8)
Vitality	61.5 (17.9)	60.8 (19.1)	60.9 (18.1)	61.6 (16.9)
Social functioning	72.3 (24.3)	73.3 (24.0)	75.5 (20.3)	69.7 (24.8)
Emotional role functioning	70.3 (38.4)	67.9 (41.2)	82.5 (29.1)	64.9 (39.2)
Mental health	67.0 (18.3)	67 (20.8)	67.2 (16.4)	62.7 (15.4)

**Table 5 cancers-14-02999-t005:** Mean scores of the scales of the SF-36 questionnaire according to sex: comparison with the general population of Medellín.

QoL Dimension	Sample of Colombian CCS (*n* = 118)	General Population Medellín * (*n* = 574)	*p*-Value **
Item	Mean (DS)	Mean (DS)	
Physical functioning	92.1 (13.5)	84.1 (24.1)	<0.001
Physical role functioning	88.1 (27.9)	77.4 (31.4)	<0.001
Bodily pain	75.9 (19.5)	74.7 (29.4)	0.58
General health	73.1 (19.6)	64.2 (22.1)	<0.001
Vitality	61.5 (17.9)	65.6 (18.1)	0.025
Social functioning	72.3 (24.3)	82.4 (23.1)	<0.001
Emotional role functioning	70.3 (38.4)	73.8 (33.2)	0.36
Mental health	67.0 (18.3)	65.7 (19.2)	0.49

* Data from Medellín taken from: García G HI, Vera G CY, Lugo A LH. Health-related quality of life (QoL) in Medellín and its metropolitan area, with the implementation of the SF-36. Revista Facultad Nacional de Salud Pública. 2013; 32(1): 26–39. Age range: 20–79 years old. ** *p*-values correspond to Student´s *t*-test for comparison of quantitative variable means.

## Data Availability

The dataset on which the study is based and the analysis code are available on request. The dataset is in Spanish.

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
