# Peer review of "Self-Reported Health Problems and Quality of Life in a Sample of Colombian Childhood Cancer Survivors: A Descriptive Cross-Sectional Study"

_cancers, 2022, doi:10.3390/cancers14122999_

Round 1

Reviewer 1 Report

Godoy-Casabuenas and de Vries have submitted their manuscript, “Perceived Long-Term Health Problems and Quality of Life in a Sample of Colombian Childhood cancer Survivors: a Descriptive Cross-Sectional Study” and do a very nice job describing the long term issues that Colombian childhood cancer survivors.  While several studies in the United States, Europe and other parts of the world have similar studies, the authors report that they are the first to report this in Columbia.  They do report on some of the more common self-reported health problems.  Overall, the paper is well written with some minor issues listed below.

Simple Summary:

  1. In the simple summary the authors use the phrase, “health problem’s perception” and in the abstract, the authors use the phrase, “health problems’ perception”.  First off, they place the apostrophe in different places, but regardless, this phrase does not make grammatical sense.  I believe the authors are trying to say, “the patients’ perception of their health problems”, and it would make more sense to say it in this way.
  2. The last sentence starting with “The aim of this study…” is a run on sentence and would be more appropriate as two separate thoughts.

Abstract:

  1. Line 24: The authors state in results, “Most of the 122…”.  This would have a greater impact if they said the exact number of 122.
  2. Line 28 (as well as throughout the rest of the paper). After the authors first use the abbreviation CCS for “Colombian childhood cancer survivors”, they do not and should not place CCS in parentheses.  It is correct the first time, but not thereafter.  For example, Line 28 should read, “Conclusion:  Perceived health problems among Columbian CCS were prevalent…”

Introduction: 

  1. Line 43: The colon is not appropriate and the sentence following should be a new sentence.
  2. Line 48: The hyphens are not appropriate and should be replaced with commas
  3. Line 63: The authors say “childhood cancer survivors” after the CCS has already been used

Materials and Methods:

  1. Page 2, Paragraph 2, starting Line 82: This paragraph is a run-on sentence and does not make sense and needs rewritten.  It would probably benefit from being broken down into more separate thoughts.
  2. Page 3, Line 91: The sentence, “…that were contacted by personal contacts of the author”.  I am not sure I understand what this means.  Who are the personal contacts of the authors – were they researchers?
  3. Page 3, Line 106: The colon is not appropriate grammatically.

Results:

  1. Page 4, Line 167: “…had already have children.” Is not grammatically correct, should be, “had already had children.”
  2. Page 5, Line 173: the word “respectively” does not need to be placed here
  3. Table 2:  The authors use the abbreviation “DoR”, but the definition of this is not on the table or in a legend
  4. Page 6, Line 196: The authors list medical specialties here, but do not discuss if any fertility specialists were seen.  As this is an important aspect of long term follow-up for adults of childhood cancer, it should be included – even if these specialists were not consulted, it would be important to add that information.
  5. Table 3:
  6. Under Reproductive system: The authors should add if fertility preservation was done in any patients
  7. Under Cardiovascular system: The authors do not list decreased cardiac function in this table which can be a late term effect of anthracycline

Discussion:

  1. Page 8, Line 240-241:  In the sentence, “…reported by many brain cancer and leukemia CCS”  It is unclear if all the reported issues are related to patients with brain cancer and leukemia, or just the  lack of concentration.
  2. Page 8-9, Line 245-249: This is a run on sentence and needs to be divided into separate thoughts
  3. Page 9, Line 259: The word “that” should be “than”
  4. Page 9, Line 266: The authors state that “This finding is in line” and “QoL among CCS to be comparable”, but their previous statement states that there were differences found, meaning that it is not in line with a study that says things are comparable. 
  5. Page 9, Line 291: The authors list the “1970s-1980s and those treated after 2000s”.  What about the 1990s-2000s?  was this just accidentally left out or not looked at?

Author Response

Response to Reviewer 1 Comments

Point 1

Simple Summary:

  1. In the simple summary, the authors use the phrase, “health problem’s perception” and in the abstract, the authors use the phrase, “health problems’ perception”.  First off, they place the apostrophe in different places, but regardless, this phrase does not make grammatical sense.  I believe the authors are trying to say, “the patients’ perception of their health problems”, and it would make more sense to say it in this way.
  2. The last sentence starting with “The aim of this study…” is a run-on sentence and would be more appropriate as two separate thoughts.

Response 1: Please provide your response for Point 1. (in red)

  1. We removed the term “health problem´s perception” and replaced it with “Perceived long-term health problems”, as it is currently in the tile.
  2. We split the last sentence into two separate thoughts.

Point 2

Abstract:

  1. Line 24: The authors state in the results, “Most of the 122…”.  This would have a greater impact if they said the exact number of 122.
  2. Line 28 (as well as throughout the rest of the paper). After the authors first use the abbreviation CCS for “Colombian childhood cancer survivors”, they do not and should not place CCS in parentheses.  It is correct the first time, but not thereafter.  For example, Line 28 should read, “Conclusion:  Perceived health problems among Columbian CCS were prevalent…

Response 2: Please provide your response for Point 2. (in red)

  1. We changed it to: “Out of the 122 CCS who participated in the study, 100% reported at least one health problem”
  2. We removed all the CCS that were in parentheses throughout the document.

Point 3

Introduction: 

  1. Line 43: The colon is not appropriate and the sentence following should be a new sentence.
  2. Line 48: The hyphens are not appropriate and should be replaced with commas
  3. Line 63: The authors say “childhood cancer survivors” after the CCS has already been used

Response 3: We made all the changes requested

Point 4

Materials and Methods:

  1. Page 2, Paragraph 2, starting Line 82: This paragraph is a run-on sentence and does not make sense and needs rewritten.  It would probably benefit from being broken down into more separate thoughts.
  2. Page 3, Line 91: The sentence, “…that were contacted by personal contacts of the author”.  I am not sure I understand what this means.  Who are the personal contacts of the authors – were they researchers?
  3. Page 3, Line 106: The colon is not appropriate grammatically.

Response 4: Please provide your response for Point 4. (in red)

  1. Page 2, Paragraph 2, starting Line 82: This paragraph is a run-on sentence and does not make sense and needs rewritten.  It would probably benefit from being broken down into more separate thoughts.
  2. Page 3, Line 91: The sentence, “…that were contacted by personal contacts of the author”.  I am not sure I understand what this means.  Who are the personal contacts of the authors – were they researchers?
  3. Page 3, Line 106: The colon is not appropriate grammatically.

 Point 5

Results:

  1. Page 4, Line 167: “…had already have children.” Is not grammatically correct, should be, “had already had children.”
  2. Page 5, Line 173: the word “respectively” does not need to be placed here
  3. Table 2:  The authors use the abbreviation “DoR”, but the definition of this is not on the table or in a legend
  4. Page 6, Line 196: The authors list medical specialties here, but do not discuss if any fertility specialists were seen.  As this is an important aspect of long-term follow-up for adults with childhood cancer, it should be included – even if these specialists were not consulted, it would be important to add that information.
  5. Table 3:
  6. Under Reproductive system: The authors should add if fertility preservation was done in any patients
  7. Under Cardiovascular system: The authors do not list decreased cardiac function in this table which can be a late-term effect of anthracycline

Response 5: Please provide your response for Point 5. (in red)

  1. Page 4, Line 202: We changed the sentence to “24 (19.7%) had children”.
  2. Page 5, Line 215: We removed the word “respectively”.
  3. Table 2: The abbreviation “DoR” refers to “Does not respond”. We added a legend under the table
  4. Page 6, Line 196: We did not include a specific question concerning visits to fertility specialists as this is usually not among the current practices for CCS in Colombia. However, we did ask participants whether they have ever been told they may have fertility issues. We included the results of this question in the above paragraph (Page 6, Line 244-245).
  5. Table 3:
  6. Under Reproductive system: We did not include a question concerning fertility preservation mainly because the focus for oncologists is to save the patient, whether the patient may have fertility issues in the future is secondary, so it is not common practice to ask about this during consultation.
  7. Under Cardiovascular system: The authors do not list decreased cardiac function in this table which can be a late-term effect of anthracycline. Although we acknowledge this is an important question, we tried to be as broad as possible in the questionnaire because as it is a self-reported questionnaire, there could be questions participants would not be able to understand. This is the reason why we did not include a question concerning decreased cardiac function and consequently, it is not reported in the table.

 Point 6

Discussion:

  1. Page 8, Line 240-241:  In the sentence, “…reported by many brain cancer and leukemia CCS”  It is unclear if all the reported issues are related to patients with brain cancer and leukemia, or just the  lack of concentration.
  2. Page 8-9, Line 245-249: This is a run on sentence and needs to be divided into separate thoughts
  3. Page 9, Line 259: The word “that” should be “than”
  4. Page 9, Line 266: The authors state that “This finding is in line” and “QoL among CCS to be comparable”, but their previous statement states that there were differences found, meaning that it is not in line with a study that says things are comparable. 
  5. Page 9, Line 291: The authors list the “1970s-1980s and those treated after 2000s”.  What about the 1990s-2000s?  was this just accidentally left out or not looked at?

Response 6: Please provide your response for Point 6. (in red) 

  1. Page 8, Line 292-293: In order to clarify the sentence, we added: “The most prevalent ones were gastritis, frequent headaches and a lack of concentration – the latter mostly reported by many brain cancer and leukemia CCS”.
  2. Page 8-9, Line 294-295: We split the sentence in two: “Several studies conducted in the United States and in Europe have described the neurocognitive problems in these survivors. These complaints are thought to be due to the brain´s vulnerability to the neurotoxicity of the therapies used to treat these types of cancer”.
  3. Page 9, Line 318: We replaced the word “that” by “than”.
  4. Page 9, Line 266: We changed the phrasing of the sentence to make it more understandable.
  5. Page 9, Line 351: We wanted to emphasize the fact that there may be difference in CCS that were treated in the early 1970s-1980s when chemotherapy and radiation therapy was just beginning in Colombia compared to those treated in more recent years, but we didn´t go in-depth as to the specific decades. We removed “the 1970s-1980s” and changed it to “in different time periods”.

Reviewer 2 Report

The current work describe health problems’ perception and quality of life among Colombian 18
adults who had cancer in their childhood or adolescence. However, the samples included is not that large enough. Moreover, the authors lack more detailed research regarding the results. Table 1 to Table 4 only described the facts, yet no comparisons. 

Author Response

Response to Reviewer 2 Comments

Response 1: We appreciate the comment. However, as mentioned in the Methods section of the paper, this is a descriptive cross-sectional study. In the limitations, we acknowledge the small sample size, the lack of sampling frame, and the use of purposeful sampling which is mainly due to the difficulties in identifying hard-to-reach and hidden populations of adults with a history of childhood cancer currently integrated into the community such as Colombian CCS. Additionally, we acknowledge that we lack a control group preventing us from performing direct comparisons with a non-cancer population and Colombian CCS. This is the reason why Table 1 and Table 4 are mainly descriptive. Yet, they hold an immense value due to the nonexistent data on this population in our country. In Table 5, however, we make a comparison of our results of QoL using SF-36 with data coming from Medellín.

Reviewer 3 Report

This manuscript presents results of a cross-sectional study of self-reported health problems and quality of life (QoL) among adult childhood cancer survivors living in Columbia. Strengths of this work include the description of important outcomes among an understudied population, the use of a self-report measure modeled off other cohort studies, and the use of the SF-36 as a validated measure of QoL.

Major Comments:

  1. I agree with the authors’ assessment that potential sampling bias in the main limitation of the current work. Although acknowledged in the Discussion, the authors should expand on how this may have influenced results. For example, by recruiting survivors who are still linked to oncologists, the sample may reflect survivors with more ongoing health problems.
  2. The authors should comment on how representative the sample is of the overall population of childhood cancer survivors in Columbia, perhaps based on incidence rates (e.g., the sample seems to have a higher proportion of survivors of leukemia and women than might be expected).
  3. The comparison sample from Medellin may be the best available, but also seems problematic in terms of basic demographics – for example, the sample was ages 20-79, while the majority of the present study sample is 18-29. Are there other published data available that might be better comparisons?
  4. The age of occurrence for many of the problems in Table 3 seem quite young compared to other cohort studies. This seems concerning and may warrant a stronger call to action in the Discussion regarding surveillance. Further, given the wide standard deviation for some of the issues, it seems that treatment era or current age may be worth considering as a factor. It would be expected that participants who are older at time of the survey and who received treatment in an earlier time period (e.g., 1970s vs. 2000s) may have received more toxic therapies and now have greater problems as they age. Consider reporting the median age of onset, rather than mean, which may address issues with outliers, if any.

Minor Comments:

  1. On p. 8, lines 219-221, “The general population had a higher score…” should be “lower score”.
  2. The final sentence of the abstract and the manuscript is “It reopens the debate on the need to carry out long-term follow-up in this population among Colombian society.” I suggest re-wording: the authors argue that survivorship has been forgotten, not that there has been any ‘debate’ about it.

Author Response

Response to Reviewer 3 Comments

Point 1

  1. I agree with the authors’ assessment that potential sampling bias in the main limitation of the current work. Although acknowledged in the Discussion, the authors should expand on how this may have influenced results. For example, by recruiting survivors who are still linked to oncologists, the sample may reflect survivors with more ongoing health problems.

Response 1: We appreciate the comment. We have added more information about this limitation in the discussion (Page 9 – Line 511-512).

Point 2

  1. The authors should comment on how representative the sample is of the overall population of childhood cancer survivors in Columbia, perhaps based on incidence rates (e.g., the sample seems to have a higher proportion of survivors of leukemia and women than might be expected).

 Response 2: We appreciate the comment. However, even though we found a higher proportion of leukemia survivors in our study, this is not representative of the whole situation in Colombia. Indeed, this is the most common type of childhood cancer in Colombia. But our sample consists of CCS coming from the most privileged urban areas of the country, those who have access to treatment and manage to survive. Unfortunately, we didn´t have any cancer survivors coming from the rural areas of the country (which is very extensive), we don´t know whether they are hard to locate or if they don´t have access to treatment and don´t survive.

Point 3

  1. The comparison sample from Medellin may be the best available, but also seems problematic in terms of basic demographics – for example, the sample was ages 20-79, while the majority of the present study sample is 18-29. Are there other published data available that might be better comparisons?

Response 3: We appreciate the comment. However, there are currently no other published data of QoL using SF-36 in our country. The sample from Medellin is the best approximation we have.

Point 4

  1. The age of occurrence for many of the problems in Table 3 seem quite young compared to other cohort studies. This seems concerning and may warrant a stronger call to action in the Discussion regarding surveillance. Further, given the wide standard deviation for some of the issues, it seems that treatment era or current age may be worth considering as a factor. It would be expected that participants who are older at time of the survey and who received treatment in an earlier time period (e.g., 1970s vs. 2000s) may have received more toxic therapies and now have greater problems as they age. Consider reporting the median age of onset, rather than mean, which may address issues with outliers, if any.

Response 4: We appreciate the comment. We agree on the importance of highlighting in the discussion the fact that most of the self-reported health problems mentioned by Colombian CCS occur at a younger age compared to other CCS cohorts. We have included a paragraph mentioning this (Page 9 Lines 720-726). Table 3: We also changed the mean age of occurrence by the median to take into account the outliers.

Point 5

Minor Comments:

  1. On p. 8, lines 219-221, “The general population had a higher score…” should be “lower score”.
  2. The final sentence of the abstract and the manuscript is “It reopens the debate on the need to carry out long-term follow-up in this population among Colombian society.” I suggest re-wording: the authors argue that survivorship has been forgotten, not that there has been any ‘debate’ about it.

Response 5:

  1. We replaced the world “higher” by “lower”.
  2. We appreciate the comment. However, in our context there has truly been a debate as to whether CCS should have a follow-up, it is not a matter of it being forgotten. Recently, follow-up for adult cancer survivors is beginning to be implemented in Colombia but there is still a long way before it is implemented in childhood cancer survivors due to the lack of organized support from healthcare systems and providers.

Reviewer 4 Report

This is a descriptive paper that adds a great deal of critically important information to the minimal literature that exists in Latinx childhood cancer survivors.  This topic intersects with childhood cancer, cancer survivorship and global health in a way that is unique and severely understudied.  This manuscript is well written and the study design is sound in all aspects.

Line 15 - 17: consider adjusting language to say: "The aim of this study is to describe the self-reported health problems and quality of life among adult -aged Colombian childhood and adolescent cancer survivors.  This will serve as the first step in characterizing this population that will support the  formulation of long-term follow-up care goals." Throughout the article is is reasonable, if supported by the author group, to replace the terms: "health problems' perception" to "self-reported health problems" which to this reader makes the message more clear.

Line 28: also, when you use the abbreviation CCS, it does not need to be in parenthesis.

Line 41: this statement: "However, the unplanned consequences of this success are not so widely known [7,8]. " while this was previously true many decades ago, it might not be applicable to the current state, as evidenced by the dates on these two references. So maybe clarify this and add some additional contextual information here such as "...in South America." or "...of this success previously was not so ...".

Line 59: adding the word "highest" after the word "second" might clarify this important fact to readers.

Line 75-77: this aim needs some clarification.  its not clear how this characterization would  serve as input for goals and who's goals would these be to begin with?  One option for clarifying this aim is as above in my first comment/suggestion.

Line 82: the word register should be registry. also consider breaking up this sentence (line 82-86) into 2 or 3 smaller sentences so the information is more easily digested by the reader.

Line 100-108: it is interesting that you did not include survivors of benign tumors that were treated with radiation since radiation has some of the highest risks for late effects.  Could you please expand on the rationale for this decision?

Line 112; I think the word "sign" should be "signing"

Line 146: you can omit the word "means" or change it to "mean scores"

Line 156-157: Can you please clarify what you mean by "Three correspond to failed..." ?  Does this mean that 3 of the 20 incomplete surveys had participants logging in multiple times to complete the questions or something else?

Line 167: this can be shortened to "Twenty-four (19.7%) had children (Table 1)."

Line 173: the word "a" can be removed here.

Line 177: the word "was" should be "were"

Line 180: is the word "referred' meant to be "reported'?

Line 253: the word "in" should be "about"

Line 259: the word "that" should be "than"

Line 323-327: this point is so critically important, would the authors consider making it in two separate sentences to clarify and highlight them further?

Author Response

Response to Reviewer 4 Comments

Point 1

Line 15 - 17: consider adjusting language to say: "The aim of this study is to describe the self-reported health problems and quality of life among adult -aged Colombian childhood and adolescent cancer survivors.  This will serve as the first step in characterizing this population that will support the  formulation of long-term follow-up care goals." Throughout the article is is reasonable, if supported by the author group, to replace the terms: "health problems' perception" to "self-reported health problems" which to this reader makes the message more clear.

Response 1: We rephrased the objective, we also changed the word “health problems perception” to “self-reported health problems” throughout the paper including the title.

Point 2

Line 28: also, when you use the abbreviation CCS, it does not need to be in parenthesis.

Response 2: We removed the parenthesis of (CCS) throughout the document.

Point 3

Line 41: this statement: "However, the unplanned consequences of this success are not so widely known [7,8]. " while this was previously true many decades ago, it might not be applicable to the current state, as evidenced by the dates on these two references. So maybe clarify this and add some additional contextual information here such as "...in South America." or "...of this success previously was not so ...".

Response 3: We appreciate your comment. We agree that the statement mentioned is a little outdated due to the substantial amount of research that has been carried out on childhood cancer survivors, mainly in high-income countries. We decided to remove this sentence. We contextualize the Latin American situation in the next paragraph (Page 2- Line 73).

Point 4

Line 59: adding the word "highest" after the word "second" might clarify this important fact to readers.

Response 4: We added the word “highest” after the word “second” as suggested.

Point 5

Line 75-77: this aim needs some clarification.  its not clear how this characterization would  serve as input for goals and who's goals would these be to begin with?  One option for clarifying this aim is as above in my first comment/suggestion.

Response 5: We rephrased the aim taking into account your first comment.

Point 6

Line 82: the word register should be registry. also consider breaking up this sentence (line 82-86) into 2 or 3 smaller sentences so the information is more easily digested by the reader.

Response 6: We changed the word “register” to “registry” as suggested. We divided the sentence into three smaller sentences to make the idea more easily transmitted.

Point 7

Line 100-108: it is interesting that you did not include survivors of benign tumors that were treated with radiation since radiation has some of the highest risks for late effects.  Could you please expand on the rationale for this decision?

Response 7: We appreciate your comment. Although we acknowledge that survivors of benign tumors treated with radiation have higher risks for late effects, we chose not to include them because of the difficulty of identifying them. The reason for this is because of the lack of a unique database of patients in the Colombian health system. In our study, we were able to identify current childhood cancer survivors mainly thanks to oncologists and cancer patient foundations who are still in contact with their patients. As benign cancer patients usually do not attend cancer patient foundations, it would have been very challenging to reach survivors of benign tumors.    

Point 8

Line 112; I think the word "sign" should be "signing"

Response 8: We replaced the word “sign” with “signing”

Point 9

Line 146: you can omit the word "means" or change it to "mean scores"

Response 9: We changed it to “mean scores”.

Point 10

Line 156-157: Can you please clarify what you mean by "Three correspond to failed..." ?  Does this mean that 3 of the 20 incomplete surveys had participants logging in multiple times to complete the questions or something else?

Response 10: We checked the number of invalid questionnaires and they were in total 10 and not 20. And yes, it means that 3 out of the 10 incomplete surveys had participants logging in multiple times to complete the questionnaire.

Point 11

Line 167: this can be shortened to "Twenty-four (19.7%) had children (Table 1)."

Response 11: We shortened the sentence as recommended.

Point 12

Line 173: the word "a" can be removed here.

Response 12: We removed the word “a”

Point 13

Line 177: the word "was" should be "were"

Response 13: We replaced the word “was” with “were”.

Point 14

Line 180: is the word "referred' meant to be "reported'?

Response 14: We replaced the word “referred” with “reported” throughout the document.

Point 15

Line 253: the word "in" should be "about"

Response 15: We replaced the word “in” by “about”.

Point 16

Line 259: the word "that" should be "than"

Response 16: We replaced the word “that” by “than”.

Point 17

Line 323-327: this point is so critically important, would the authors consider making it in two separate sentences to clarify and highlight them further?

Response 17: We split the sentence in two in order to clarify and highlight them further.
